# Exploring Patient Trust in Healthcare Provider Influenza Vaccine Information and Recommendations in a Medically Underserved Area of Washington State

**DOI:** 10.3390/vaccines13050505

**Published:** 2025-05-10

**Authors:** Damianne Brand, Megan Giruzzi, Nick Giruzzi, Kavya Vaitla, Rose Krebill-Prather, Juliet Dang, Kimberly McKeirnan

**Affiliations:** 1Pharmacotherapy Department, Yakima Campus, Washington State University, Yakima, WA 98901, USA; dbrand@wsu.edu (D.B.); megan.giruzzi@wsu.edu (M.G.); nicholas.giruzzi@wsu.edu (N.G.); 2Pharmacotherapy Department, Spokane Campus, Washington State University, Spokane, WA 99202, USA; kvaitla2@gmail.com; 3Social and Economic Sciences Research Center, Pullman Campus, Washington State University, Pullman, WA 99164, USA; krebill@wsu.edu; 4CSL Seqirus, Summit, NJ 07901, USA; juliet.dang@seqirus.com

**Keywords:** influenza vaccination, vaccine safety, patient–provider trust, underserved populations, Hispanic community

## Abstract

**Background/Objective**: Patients have historically trusted healthcare providers to be a reliable source of health information. However, with the recent pandemic and subsequent recovery, understanding and developing patients’ trust has become even more important, especially regarding vaccine acceptance. The objective of this work is to explore the current level of trust that rural patients have in their healthcare providers concerning influenza vaccination and related recommendations and its impact on vaccine uptake in a rural county in Washington State. **Methods**: An anonymous survey was conducted by a survey research center using a random sampling of 3000 addresses for people living in Yakima County in Washington State. Yakima County has a high percentage of people who identify as Hispanic or Latino/a and is a medically underserved area. The survey was designed to evaluate factors influencing the decision to be vaccinated against influenza and the level of trust in information from healthcare providers. **Results**: Results showed that participants who had been vaccinated against influenza in the previous five years were more likely to trust the advice of their primary care provider (*p* < 0.001), specialty care provider (*p* < 0.001), pharmacist (*p* = 0.02), and nurse (*p* = 0.002). People who were not vaccinated against influenza in the last five years were statistically more likely to report that a recommendation from a healthcare provider would not make a difference in their decision (*p* < 0.001). People who were vaccinated were more likely to utilize healthcare providers as a source of information about the influenza vaccine (*p* < 0.001) and people who were unvaccinated were more likely to use their own personal research as a trusted information source (*p* = 0.04). **Conclusions**: Healthcare providers continue to be well regarded and trusted by their patients, especially in rurally located counties, though work still needs to be carried out around influenza vaccination importance messaging. This work identified that all healthcare providers need to work collaboratively to reinforce vaccination guideline recommendations and to both provide education and continue successful access-to-vaccination strategies to promote influenza prevention.

## 1. Introduction

Healthcare providers (HCPs) have long enjoyed a substantial level of trust among those they serve, with physicians, pharmacists, and nurses consistently scoring in the top 10 most trusted professions in annual Gallup Polls even as recently as January of 2024 [1]. When asked more specifically about trust in their provider, patients stated they had greater trust in their HCP than they had in the healthcare system. Ironically, when providers were asked about trust, many stated that they understood the importance of building trust with their patients but admitted to not always engaging in trust-building behaviors [2].

In recent years, trust in HCPs has been decreasing [3,4]. Factors influencing patient trust in the healthcare system since the Coronavirus Disease 2019 (COVID-19) pandemic are complex and difficult to measure [5]. One issue is healthcare system change and the expansion of managed care causing a decrease in time spent together in appointments [6]. Trust is an integral part of the patient–provider relationship and is one of the common drivers of patient satisfaction, which is often used as a measure of quality in healthcare and in turn can affect clinical outcomes [7]. The trust developed and maintained by HCPs is integral in ensuring that patients seek out care, feel comfortable with disclosure, and take recommendations or make necessary behavior changes suggested by their providers [8]. This trust can be built through open communication between the patient and their provider, allowing the patient to experience greater autonomy in treatment-based decision making [9]. Confidence, according to the World Health Organization 3C Model of Vaccine Hesitancy, is built through developing patient trust in vaccine safety and effectiveness, trust in HCPs and services, and trust in the decisions of vaccination policy makers [10]. Addressing Confidence, as well as the other 3Cs (Complacency and Convenience) can aid HCPs in addressing vaccine hesitancy with their patients [11].

Patient satisfaction and subsequent outcomes were negatively affected by the COVID-19 pandemic [3,4]. Satisfaction and in-person care were extremely limited, requiring providers to find new ways to continue contact with their patients through cumbersome and complicated means disrupting routine clinical interventions and visits [3,4]. Though trust in providers remained strong as many HCPs continued to serve on the front lines, the patient–provider relationship diminished. As the pandemic progressed, an environment emerged that was rich in misinformation about the state of care from long-trusted media outlets, health authorities, and social media. Additionally, distancing not only physically but through the wearing of personal protective equipment created physical barriers affecting those who were hearing impaired and made the recognition of visual cues important to conversation difficult to distinguish. Some of these barriers included an inability to use technology, a lack of assistance from others, a lack of literacy, a lack of resources (e.g., video camera, smart phone), and hearing impairment. These barriers affected patients nationwide but had the greatest impact on our elderly population [9]. The COVID-19 pandemic brought to light the underlying inequities associated with healthcare, including disproportionate access to testing and treatment [12]. Additionally, the changing sociopolitical environment in healthcare caused a divide between beliefs and scientific facts, and the increased access to information technology allowed patients to feel empowered to make “informed” decisions based on their own “research” [9].

Influenza vaccination is a yearly topic of discussion at HCP offices. Influenza vaccination remains the number one preventative approach due to its effectiveness at reducing the morbidity, mortality, and spread of the disease [13]. Unfortunately, influenza vaccination rates have been affected by this shift in HCP trust. For the 2020–2021 influenza season, vaccination rates were at an all-time high, with 50.2% of adults aged 18 and older receiving the influenza vaccine [14]. Since this, peak vaccination rates have dropped to pre-pandemic levels, with only 45.4% of adults being vaccinated for the 2021–2022 influenza season [15]. If vaccination rates continue to decline due to an emerging vaccine hesitance, we may not be able to achieve the levels of immunity necessary to control influenza in the community.

Many studies have investigated the cause of vaccine hesitancy and determined that the three main causes are vaccine-related, health system-related, and related to individuals’ social attributes [16]. These factors are further influenced by various psychological, sociocultural, political, and media-related factors [17]. The responsibility for many interventions to help overcome these factors and improve vaccination uptake falls on HCPs. These interventions are adding to the increased pressures that HCPs are already facing with ever-changing guidelines and insurance-based outcome metrics. Additionally, HCPs are experiencing increased patient questions about vaccinations. Providers are faced with resource shortages and high patient volumes and may not have time to provide the education necessary to address different attitudes/beliefs about vaccinations [18]. All of these factors are likely contributing to the decreased levels of trust in HCPs.

The goal of this quantitative research was to explore the perceived trust that patients in a rural community of Washington State have in the HCPs that serve them in relation to influenza vaccination education and prevention and subsequent impacts on vaccination.

## 2. Materials and Methods

### 2.1. Study and Research Tool Design

Faculty from the Washington State University College of Pharmacy and Pharmaceutical Sciences (WSU) designed a survey after performing a literature search on reasons for hesitancy and barriers in influenza vaccination [19,20,21,22,23,24]. The Social Economic Sciences Research Center (SESRC) team from WSU assisted with survey language, organization, and dissemination. The survey itself focused on assessing influenza vaccination perspectives and participation in a rural community, including the impact of trust in HCP recommendations discussed in this article. After review and feedback by two vaccination industry experts, the finalized survey included 64 questions, 17 of which are discussed within this manuscript and included in Table 1. These questions are re-numbered from the original 64 for clarity and continuity. The remaining results will be reported in separate publications [25].

### 2.2. Participants and Data Collection

The SESRC team managed and distributed both an electronic and a paper version of the survey. The team purchased a random sample of 3000 mailing addresses, excluding residents’ names, within Yakima County, Washington, from the United States Postal Service Computerized Delivery Sequence File. The survey and information sent to potential respondents were in English. Conducting the survey in multiple languages would have been ideal but was also prohibited by cost. The SESRC also noted that historically, few respondents use the Spanish translation of materials to complete surveys since many are bilingual. The 3000 potential respondents were contacted up to five times to request their participation. Initially, each household was mailed a notification letter with a one-dollar incentive and link to the survey. One week later, a paper survey packet with a web link was mailed to each household, and one week after that, a postcard reminder was sent. A replacement questionnaire was mailed one week after the postcard, and a final reminder letter was sent two weeks later. On each occasion, the potential respondents received a statement indicating that the survey was voluntary and confidential, and results would be aggregated for anonymity. After the five consecutive postal mailings, 500 surveys were obtained, representing an 18.3% response rate +/−4.4% due to sample error. These survey procedures were developed using the American Association of Public Opinion Research survey best practices guidelines [26].

### 2.3. Data Analysis

The survey results were aggregated by the SESRC team. The survey dataset was exported into SPSS version 29 for analysis and weighted by age, gender, and ethnicity. Weighting aligned respondent representation with characteristics of the general population living in Yakima County, Washington, using data from the United States Census [27]. Categorical variables were analyzed using descriptive statistics (frequencies) for the demographics data. During analysis, participants were stratified into two categories based on their response to Question 7 regarding influenza vaccination in the past five years: respondents who had never been vaccinated were categorized as “unvaccinated” and participants who had been vaccinated against influenza at least once during the previous five years were considered “vaccinated” for the purpose of the analysis. The chi-squared test was utilized to compare data in the vaccinated and unvaccinated populations.

### 2.4. Ethical Approval

This study was granted approval, as it was found that this research satisfied the criteria for Exempt Research (IRB#19877-001) by Washington State University Investigational Review Board (IRB) and was conducted following university research governance procedures.

## 3. Results

### 3.1. Recent Influenza Vaccination Status

Out of 500 respondents, 420 were included in the analysis based on survey completeness. The demographics of the 420 study participants are included in Table 2. In response to Question 7, 41.1% of respondents indicated that they had been vaccinated against influenza every year within the past five years and 26.4% had received the vaccine at least once within the past five years; however, 29.1% indicated they had not had the flu vaccine at all within the past five years (2018–2022) and 3% indicated that they were unsure. Overall, 70.9% of respondents had been vaccinated at least once within five years and 29.1% had not been vaccinated at all in the time period. Vaccination status in the last five years was utilized to organize the results for the subsequent questions.

### 3.2. Participant Trust in Healthcare Provider Influenza Vaccine Recommendations

Participants were asked, “Has a healthcare provider recommended getting vaccinated within the past five years?” (Question 8). A chi-squared test was used to determine whether there was a significant difference between the proportion of vaccinated and unvaccinated respondents whose HCP recommended being vaccinated against influenza. The results indicated that 76.6% of vaccinated respondents versus 49.0% of unvaccinated respondents had received a recommendation from an HCP to receive the influenza vaccine in the last five years (x^2^ df = 1, N = 408, x^2^ = 28.464, *p* < 0.001), as shown in Figure 1.

Among those who reported that an HCP had recommended an influenza vaccine in the last five years, the frequency of recommendations received is shown in Table 3. People who were vaccinated report receiving recommendations more frequently than people who were not vaccinated (x^2^ df = 1, N = 291, x^2^ = 38.953, *p* < 0.001).

Participants who reported that an HCP had recommended an influenza vaccine in the last five years were asked which type of provider(s) made the recommendation (Question 10). The results are shown in Table 4. Significantly more people who were vaccinated reported receiving a recommendation from a primary care provider (x^2^ df = 1, N = 281, x^2^ = 13.942, *p* < 0.001), a specialty care provider, such as a cardiologist or endocrinologist (x^2^ df = 1, N = 245, x^2^ = 28.845, *p* < 0.001), or a pharmacist (x^2^ df = 1, N = 252, x^2^ = 9.714, *p* = 0.002). No significant difference was found regarding recommendations from a nurse.

Survey participants were asked whether they had confidence in the advice that they received from HCPs (Question 11). The results showed that participants who had been vaccinated against influenza in the previous five years were significantly more likely to trust the advice of their primary care provider (*p* < 0.001), specialty care provider (*p* < 0.001), pharmacist (*p* = 0.02), and nurse (*p* = 0.002). The participant responses are shown in Table 5.

Survey participants were also asked whether a recommendation from a trusted HCP would have made a difference in their decision to receive the flu vaccine (Question 12). People who were not vaccinated against influenza in the last five years were more likely to report that a recommendation from an HCP would not make a difference in their decision (x^2^ df = 1, N = 421, x^2^ = 102.045, *p* < 0.001), as shown in Table 6.

### 3.3. Participant Sources of Information About Influenza Vaccine

Survey participants were asked which sources of information they used to obtain information about the influenza vaccine (Question 13). The results showed that people who were vaccinated were more likely to utilize HCPs as a source of information about the influenza vaccine (*p* < 0.001) and that people who were unvaccinated were more likely to use their own personal research as a trusted information source (*p* = 0.04). There was no statistical difference between the use of mass media (*p* = 0.627), social media (*p* = 0.547), or information from a school or college (*p* = 0.245). The responses are shown in Table 7.

Participants were also asked to rate their level of confidence with the information that they had received from HCPs about the influenza vaccine. Participants who had been vaccinated against influenza in the last five years were more likely to believe that they had received trustworthy information about the flu vaccine (*p* < 0.001), that the vaccine was safe (*p* < 0.001), and that they understood how the flu vaccine worked (*p* = 0.003). The responses are shown in Table 8.

A chi-squared test was used to determine whether there was a significant difference between the proportion of vaccinated and unvaccinated respondents pertaining to how having an understanding how vaccines work would impact interest in being vaccinated. We observed that 53% percent of vaccinated respondents compared with 29% of unvaccinated respondents reported that having a deeper understanding of how vaccines worked would make a difference in their interest in being vaccinated (24.134, *p* < 0.001). The results are shown in Figure 2.

## 4. Discussion

To mitigate vaccine hesitancy, HCPs must start by building trust and meeting patients where they are in their understanding of influenza vaccination. Trust in HCP recommendations and the professions themselves has fortunately only mildly decreased over the last few years, even in the wake of the most recent pandemic. Throughout the survey, there were positive patient–provider relationships but also much room for improvement in the areas of building trust, a better understanding of patient education sources, and routine communication regarding vaccinations. It was found that survey responders were more likely to receive the influenza vaccine if they trusted the recommendations from their HCP. Survey respondents stated they mostly or strongly trusted the advice of specialty care providers (cardiologists, pulmonologist, etc.) at 89.6% and their primary care provider at 85.8%. Primary care providers included physicians, nurse practitioners, and physician assistants and in most cases represented the general practitioner that they would see for yearly check-ups, disease state management, and seasonal illnesses. Both nurses and pharmacists scored similarly, with approximately 83% of respondents stating that they had trust in these professions. This result was interesting as pharmacists are the most accessible HCPs, with patients visiting their community pharmacists nearly twice as often as their general practitioner, so it seems that potentially, greater access/exposure to an HCP may not contribute to levels of trust [28]. With increased accessibility and trust, pharmacist-provided vaccinations can positively affect cost and contribute to a reduction in patient barriers in underserved and rural communities. However, as with all HCPs, they are being hindered by administrative barriers such as financial resources, medication short stocks, inadequate staffing, and higher volumes of prescriptions dispensed. None of these barriers allow for appropriate time to serve the patient in a timely and in-depth manner [29].

When participants were asked if they had received a provider recommendation for flu vaccination by their HCP in the last 5 years, most (396/521 (71%)) answered yes. This was found to be primarily offered during flu season when this type of vaccination was front of mind vs. during routine visits. There is good support in the research showing that provider recommendations have a direct impact on vaccination rates [30]. This is reflected in the data, with 58.4% of participants stating that a recommendation by a provider alone would push them to undergo a vaccination. This repeated support of vaccination at each visit could help solidify the need for the patient to ensure that it is carried out.

In contrast to recommendations, when participants were asked, “How often have HCP been offering you a flu vaccination over the past 5 years”, meaning providing it there in their office or place of business, physicians were the highest, with 351, followed by pharmacists scoring 115 and nurses scoring 100. This shows that even though pharmacists are both trusted and the most accessible HCPs, there is still the opportunity to improve pharmacists’ communication and patients’ understanding of the pharmacist’s role and offerings to the community to potentially increase this impact [31,32]. It is also important to consider that patients interacting with physicians and nurses are often at an appointment focused on a separate health problem. The topic of vaccination may not be prioritized during these visits. Patients experiencing stress related to new diagnoses or treatment plans may also be overwhelmed and less able to listen to and understand vaccination recommendations.

With these levels of trust in mind, the survey participants were asked “Where do you get information about flu vaccines?” Most positive responders gave their HCP as their primary locum of information at 419 out of 480 (80.6%). It has been found that provider recommendations of vaccines, especially with concurrent education on why they are important, represent one of the strongest predictors that the patient will follow through with receiving the vaccination [33]. Though it is positive that HCPs still have this level of influence, there should be equal vigilance associated with accurate information reaching the patient. Gaps in knowledge around vaccine type availability and staying up to date with vaccination schedules and guidelines, indications, and side effects are difficult to address as many providers are struggling with increasing workloads in addition to the influx of new innovations and updated treatment information [34]. Additionally, survey respondents reported obtaining their information on the influenza vaccines by conducting their own research at 245/429 (47%) and through standard media including television/radio at 210/453 (40.4%). Conducting individual research (with the help of Google and ChatGPT) and relying on social media have become popular behaviors for gaining information regarding vaccinations over the last decade. These responses reflect what has been witnessed by HCPs as of late around internet accessibility and its use by younger generations and the more traditional commentary supplied by the nightly news consumed by older generations. Both the polarization of media and the increased accessibility of the internet for learning and delving deep to search for answers have allowed beliefs and misinformation to be given the same status as more rigorous traditional research and medical organizations [35]. HCPs can help mitigate this misinformation by not only staying up to date with vaccination guidelines and available offerings but also ensuring that the language used to promote and educate vaccination is positively framed. An example of positive vs. negative language could be “side effects are unlikely” vs. “there is a chance of side effects [36,37].

Regardless of how they receive the information, the majority of respondents reported being mostly or very confident (82.2%) in the vaccine information that they were receiving. Interestingly, this confidence in the information received did not align with the confidence in their own knowledge on how the vaccine operated (76.6%). There also did not seem to be a connection with patients’ understanding on how the vaccine worked and their interest in receiving one. Fifty-one percent stated that it would probably or definitely not increase vaccination interest, with only 35% stating that it would. A small portion just answered that they “didn’t know”. Whether the participant understood how the vaccine worked or not, regarding participants’ confidence in the flu vaccine’s safety, most (80.5%) believed that it was mostly or very safe. This response was positive for future vaccine offerings and showed both the resiliency of the patient against misinformation and the potential positive outcomes of vaccine advocacy by HCPs and medical organizations.

It is important to note that there are limitations to this work; primarily, the results reflect the perspectives of a single rural county/community in Washington State and may not be representative of other areas within the state, other states, or other countries. Additional research is needed to determine whether similar results would be found in other states and urban areas. However, the results were in line with national reporting regarding trust in HCPs and influenza vaccination acceptance. Additionally, the survey results are self-reported, introducing the opportunity for recall bias due to respondents either remembering or reporting information inaccurately.

There are also limitations related to language and potential understanding. Participant levels of education and health literacy were not evaluated, so it is possible that respondents may not have fully understood the questions. This survey was only conducted in English and may not have fully reflected opinions of members of the community who did not speak English. Ideally, the survey would have been conducted in multiple languages, but this was not an option for this project due to the cost of translating the survey into other languages.

## 5. Conclusions

A patient’s trust in their HCP is a contributing factor in understanding the importance of routine influenza vaccination. Confidence in HCP recommendations can educate patients and aid them in making healthcare decisions that are in their best interest. HCPs continue to be well regarded and trusted by their patients, especially in rurally located counties, though work still needs to be carried out around influenza vaccination importance messaging. Overall, the responses from survey participants suggest that HCPs need to continue to improve communication regarding vaccination guideline recommendations, and this may be an area of interprofessional collaboration when providing both education and access to vaccination to promote influenza prevention.

## Figures and Tables

**Figure 1 vaccines-13-00505-f001:**
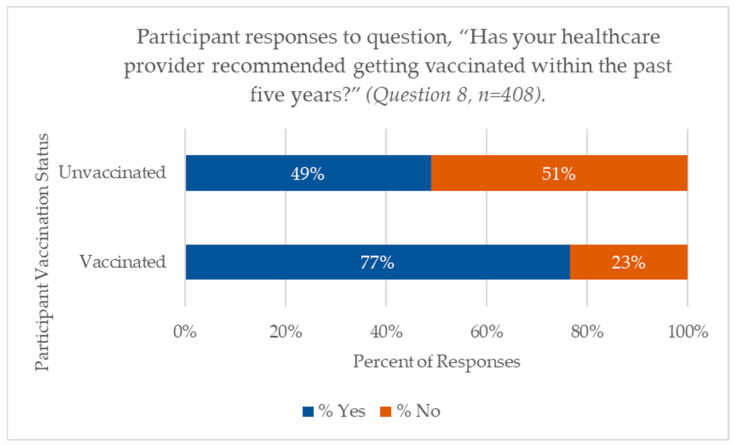
Participant responses to question, “Has your healthcare provider recommended getting vaccinated within the past five years?” (Question 8, n = 408). Unvaccinated patients were those who reported not receiving an influenza vaccination in the last five years in response to Question 7.

**Figure 2 vaccines-13-00505-f002:**
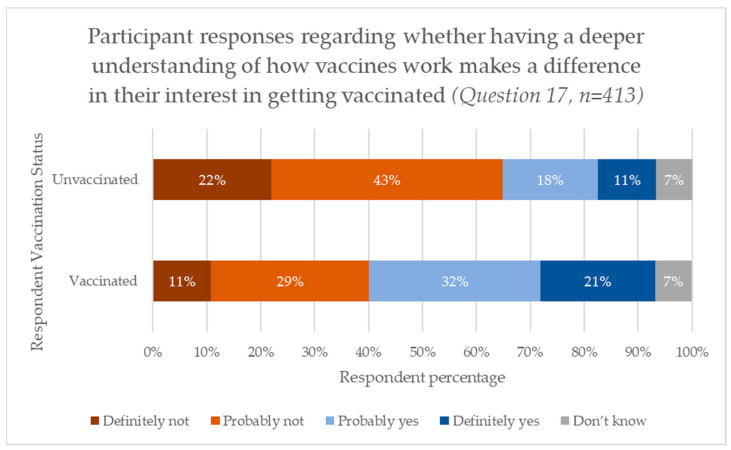
Participant responses regarding whether having a deeper understanding of how vaccines work makes a difference in their interest in being vaccinated (Question 17, n = 413). Unvaccinated patients are those who reported not receiving an influenza vaccination in the last five years in response to Question 7.

**Table 1 vaccines-13-00505-t001:** Survey questions.

Number	Survey Question	Domain
1	Have you lived in Yakima County for the past 12 months or longer?	Inclusion criteria
2	Are you age 18 or older?	
3	Have you ever received an influenza vaccination?	
4	Which of the following is your age group?	Demographics
5	Which gender do you identify as?	
6	Please indicate your ethnicity: Latino/Latina/Latinx/Hispanic; Non-Hispanic.	
7	How often in the past five years (2018–2022) have you been vaccinated for the flu?	Vaccination status
8	Has your healthcare provider recommended for you to get vaccinated against the flu within the last 5 years (2018–2022)?	Trust in provider recommendations
9	(If yes to previous) How often have health care providers been offering you a flu vaccination over the past 5 years?	
10	(If yes to Question 9) Which healthcare provider(s) recommended getting a flu vaccine?	
11	How much do you trust the advice of each of the following healthcare providers?	
12	Would a strong recommendation from a trusted healthcare provider influence your decision to get vaccinated?	
13	Where do you get information about flu vaccine?	Information about influenza vaccine
14	How confident are you that you have received trustworthy information about the flu vaccine?	
15	How confident are you that the flu vaccine is safe?	
16	How confident are you in your knowledge of how the flu vaccine works?	
17	Would having a deeper understanding of how vaccines work make a different in your interest in getting vaccinated?	

**Table 2 vaccines-13-00505-t002:** Demographics of survey respondents included in analysis (*n* = 420).

Demographic	Respondents, N (%)
Ethnicity (Question 4)	
Hispanic/Latino/Latina/Latinx	180 (43%)
Non-Hispanic/Latino/Latina/Latinx	240 (57%)
Gender (Question 5)	
Male	217 (52%)
Female	203 (48%)
Non-binary or other	0 (0%)
Age (Question 6)	
18–24 years	46 (11%)
25–34 years	85 (20%)
35–44 years	71 (17%)
45–54 years	68 (16%)
55–64 years	68 (16%)
65–79 years	65 (15%)
80 years or older	17 (4%)

**Table 3 vaccines-13-00505-t003:** Participant responses to Question 9: How often have healthcare providers been offering you a flu vaccination over the past 5 years? (n = 296).

Population	Frequency of Provider Recommendation			
Population	At Every Visit	Annually During Flu Season	Occasionally but Not Every Year	Never
Unvaccinated *	18.3%	38.9%	30.5%	12.3%
Vaccinated	18.6%	72.6%	6.3%	2.5%

* Unvaccinated patients were those who reported not receiving an influenza vaccination in the last five years in response to Question 7.

**Table 4 vaccines-13-00505-t004:** Participant responses to Question 10: Which healthcare provider(s) recommended getting a flu vaccine? (n = 296).

Healthcare Provider Type	Population	Recommended Influenza Vaccine?		Chi-Squared Likelihood Ratio	df	*p*-Value
		Yes	No			
Primary care provider (n = 281)	Unvaccinated *	81.3%	18.7%	13.942	1	<0.001
	Vaccinated	96.4%	3.6%			
Specialty care provider (n = 245)	Unvaccinated *	3.2%	96.8%	28.845	1	<0.001
	Vaccinated	34.3%	65.7%			
Pharmacist (n = 252)	Unvaccinated *	18.1%	81.9%	9.714	1	0.002
	Vaccinated	39.4%	60.6%			
Nurse (n = 245)	Unvaccinated *	37.0%	63.0%	0.149	1	0.6997
	Vaccinated	39.8%	60.2%			

* Unvaccinated patients were those who reported not receiving an influenza vaccination in the last five years in response to Question 7.

**Table 5 vaccines-13-00505-t005:** Respondent trust in healthcare providers (Question 11).

Healthcare Provider	Patient Population *	Strongly Trust	Mostly Trust	Only Trust a Little	Do Not Trust at All
Primary care provider	Unvaccinated *	30.1%	52.7%	14.1%	2.1%
	Vaccinated	49.4%	41.7%	5.5%	3.4%
Specialty care provider	Unvaccinated *	37.6%	42.9%	16.1%	3.4%
	Vaccinated	55.4%	34.6%	4.5%	5.5%
Pharmacist	Unvaccinated *	21.4%	50.5%	23.8%	4.3%
	Vaccinated	36.4%	44.6%	15.4%	3.6%
Nurse	Unvaccinated *	27.6%	42.9%	27.4%	2.1%
	Vaccinated	36.0%	48.2%	11.7%	4.1%

* Unvaccinated patients were those who reported not receiving an influenza vaccination in the last five years in response to Question 7.

**Table 6 vaccines-13-00505-t006:** Influence a recommendation from a trusted healthcare provider would have in decision to receive the flu vaccine (Question 12, n = 421).

	Definitely Yes	Probably Yes	Probably Not	Definitely Not	Unsure	Chi-Squared Likelihood Ratio	df	*p*-Value
Unvaccinated *	3.1%	21.3%	47.4%	21.5%	6.7%	102.045	4	<0.001
Vaccinated	35.0%	38.0%	17.1%	5.5%	4.4%			

* Unvaccinated patients were those who reported not receiving an influenza vaccination in the last five years in response to Question 7.

**Table 7 vaccines-13-00505-t007:** Sources of information about influenza vaccine as reported by survey respondents (Question 13).

Information Source	Patient Population *	Yes	No	Chi-Squared Likelihood Ratio	df	*p*-Value
Healthcare provider (n = 397)	Unvaccinated *	72.2%	27.8%	11.971	1	0.0005
	Vaccinated	87.3%	12.7%			
Media such as TV, radio, newspaper (n = 366)	Unvaccinated *	52.7%	47.3%	0.237	1	0.6268
	Vaccinated	49.9%	50.1%			
Social media (n = 356)	Unvaccinated *	22.8%	77.2%	0.3624	1	0.5472
	Vaccinated	25.9%	74.1%			
School or college (n = 360)	Unvaccinated *	23.6%	76.4%	0.0492	1	0.2450
	Vaccinated	24.7%	75.3%			
Your own personal research (n = 375)	Unvaccinated *	68.8%	31.2%	4.3814	1	0.0363
	Vaccinated	57.2%	42.8%			

* Unvaccinated patients are those who reported not receiving an influenza vaccination in the last five years in response to Question 7.

**Table 8 vaccines-13-00505-t008:** Respondent confidence about influenza vaccine (Questions 14–16).

Question	Patient Population *	Very Confident	Mostly Confident	Not Very Confident	Not at All Confident
How confident are you that you have received trustworthy information about flu vaccines? (Question 14, n = 413)	Unvaccinated *	14.7%	51.0%	19.1%	15.2%
	Vaccinated	38.5%	45.1%	11.7%	4.7%
How confident are you that the flu vaccine is safe? (Question 15, n = 415)	Unvaccinated *	14.9%	36.3%	31.3%	17.5%
	Vaccinated	46.0%	43.2%	5.4%	5.4%
How confident are you that you understand how the flu vaccine works? (Question 16, n = 413)	Unvaccinated *	14.5%	41.9%	32.5%	11.1%
	Vaccinated	27.1%	49.0%	19.5%	4.4%

* Unvaccinated patients are those who reported not receiving an influenza vaccination in the last five years in response to Question 7.

## Data Availability

The datasets presented in this article are not readily available because the data are part of an ongoing study. Requests to access the datasets should be directed toward the corresponding author.

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
