# Peer review of "Exploring Patient Trust in Healthcare Provider Influenza Vaccine Information and Recommendations in a Medically Underserved Area of Washington State"

_vaccines, 2025, doi:10.3390/vaccines13050505_

Round 1

Reviewer 1 Report

Comments and Suggestions for Authors

This is a clearly presented report on a subset of questions, from a larger survey sent electronically and on paper to 3000 households.  The response rate of 18% is low, but seems to include a diversity of persons living in Yakima County.  The finding that people who did and did not receive flu vaccine overall still respected their providers is reassuring. It is not surprising that those who were vaccinated were more likely to report a recommendation from a provider or pharmacist.  Is it possible that the lower rates for the unvaccinated represent that those people did not listen or did not remember the recommendation?

I have two comments about your discussion and conclusions.

  1. I am not sure you can say that your work identified that all healthcare providers need to work collaboratively to reinforce vaccination recommendations.  While that is likely true, and a great idea, please specify how your work reinforces that; or else make it an observation or recommendation from you, and not claim that your work reinforces this.
  2. A thought regarding pharmacists. You say in first paragraph of discussion that you are surprised that only 83% of respondents stated trust in pharmacists.  I believe that is actually pretty high, but even if lower would not be surprising if the pharmacists in Yakima County are in big company pharmacies like Walgreens and CVS.  In the past, the local pharmacist would know his or her clients; but with the large chains, pharmacists likely work in larger teams and have shifts, making it unlikely that they can get to know clients.

Finally, on lines 125-127, you explain why the survey was only in English, despite the relatively large Hispanic population in Yakima County.  You say "historically few respondents use the Spanish translation of materials to complete surveys".  Do you mean, they use the English versions? Or do they not respond?  If the latter, this is a limitation, you may have a portion of your population not represented, and you might consider that as one reason for the low response rate.  In general in this day and age in the US a Spanish language version of a survey is a good idea.  Perhaps you can give statistics for Yakima County and show how well your respondents match by ethnicity and age group.

Author Response

Please see the attached table.

Reviewer 2 Report

Comments and Suggestions for Authors

This study fills a gap in research on influenza vaccine confidence in rural medically underserved areas, but needs to improve the methodological rigour, interpretive power of the results, and practical guidance value through the above revisions. It is recommended that it be revised and resubmitted for peer review. Specifically as follows:

1. The sampling method was based on a random sample of postal addresses, but did not specify whether it included people with no fixed address or non-English speakers. The survey was conducted in English only, which may have resulted in an under-representative sample.

2. The application scenarios of the Chi-square test and Fisher's exact test were not clearly distinguished (e.g. it was not stated when the latter was used due to an expected frequency <5). In addition, there was no mention of whether correction for multiple comparisons (e.g. Bonferroni correction) was used, which may increase the risk of type I error.

3. some of the citations of the effects of COVID-19 on doctor-patient trust (e.g. references [3][4][8]) were not updated to include studies after 2023. In addition, the field of vaccine trust lacks key theories (e.g., the "3C model of vaccine hesitancy" or the health belief model).

4. Limitations of the study, such as the sample being limited to a single rural county (Yakima County), the potential for recall bias (reliance on self-reported vaccination history), and the lack of control for confounding variables (e.g., education level, health literacy) were not adequately discussed.

5.The conclusion mentions that "collaboration and education strategies need to be strengthened", but does not specify how this might be done (e.g., how to improve communication skills of health workers or optimise vaccine accessibility).

6.The labelling of Figure 1 and tables was not clear enough, and there was a mismatch between figures and text citations (e.g. inadequate interpretation of Tables 3 and 4).

7.Too much is plagiarised

Author Response

Please see the attached table.

Reviewer 3 Report

Comments and Suggestions for Authors
  1. In the text, the abbreviations should be interpreted with their full spellings in the first place.
  2. In Table 3, would the authors please check the frequency of provider recommendation for “Vaccinated" is right? 18.5%+72.6%+6.2%+2.4%=99.7%, but not 100%. Other Tables (5, 6, 8) should also be checked.
  3. The degree of freedom was spelled as DF in some places, but df in other places. Would the authors please unify the spelling, the same for P value.

Author Response

Thank you for the helpful comments. Here are our specific responses:

  1. In the text, the abbreviations should be interpreted with their full spellings in the first place.

Thank you for the feedback. We have defined COVID-19 at first use and changed “COVID” to “COVID-19” throughout. We also changed US to United States. HCP, SESRC, and WSU were all previously defined at first use.  

  1. In Table 3, would the authors please check the frequency of provider recommendation for “Vaccinated" is right? 18.5%+72.6%+6.2%+2.4%=99.7%, but not 100%. Other Tables (5, 6, 8) should also be checked.

We appreciate the reviewer pointing this out. Table 3, 5, 6, and 8 were checked and fixed.

  1. The degree of freedom was spelled as DF in some places, but df in other places. Would the authors please unify the spelling, the same for P value.

Changed to lowercase df and p throughout.

Round 2

Reviewer 2 Report

Comments and Suggestions for Authors

The authors' response to the reviewers' comments is generally careful and thorough, and key points are largely revised or explained. The paper has a good structure, the data analysis has been stringent, and the discussion and conclusions have been built on the results. Although there are still some minor issues in detail, these do not affect the overall merit and it is recommended that the Editorial Board accepts the paper by synthesising the opinions of all reviewers.

Reviewer 3 Report

Comments and Suggestions for Authors

The issues have been addressed, I have no more suggestions.